# Muscle strength gains per week are higher in the lower-body than the upper-body in resistance training experienced healthy young women—A systematic review with meta-analysis

Roger Jung[1]*, Sebastian Gehlert[1], Stephan Geisler[2], Eduard Isenmann[2,3], Julia Eyre[2], Christoph Zinner[4]

1 Department of Biosciences of Sport Science, Institute of Sport Science, University of Hildesheim, Hildesheim, Germany, 2 Fitness and Health, IST University of Applied Sciences, Duesseldorf, Germany, 3 Department for Molecular and Cellular Sports Medicine, Institute for Cardiovascular Research and Sports Medicine, German Sport University Cologne, Cologne, Germany, 4 Department of Sport, University of Applied Sciences for Police and Administration of Hesse, Wiesbaden, Germany

* rcr_jung@gmx.net

**Data Availability Statement:** All relevant data are within the paper and its Supporting Information files.

## Abstract

### Background

Women are underrepresented in resistance exercise-related studies. To date only one meta-analysis provides concrete training recommendations for muscle strength gains through resistance training in eumenorrhoeic women.

### Objective

This review aims to identify research gaps to advance future study in this area to expand the knowledge concerning resistance exercise-induced strength gains in women and to provide guidelines on the number of repetitions per set and the training frequency per week to enhance maximal muscle strength.

### Methods

The electronic databases PubMed and Web of Science were searched using a comprehensive list of relevant terms. After checking for exclusion criteria, 31 studies could be included in the final analysis using data from 621 subjects. From these data sets, the ideal number of repetitions per set and also the training frequency per week were analyzed.

### Results

In the lower body, the largest gains were achieved with 1 to 6 repetitions (17.4% 1RM increase). For lower-body exercises, the highest gains were achieved with 13 to 20 repetitions (8.7% 1RM increase). The lower body should be trained two times a week (8.5% 1RM increase). The upper body should be trained two (5.2% 1RM increase) to three times (4.5% 1RM increase) a week.

**Funding:** The authors received no specific funding for this work.

**Competing interests:** The authors have declared that no competing interests exist.

## Conclusion

Women can increase their 1RM by 7.2% per week in the upper body and by 5.2% per week in the lower-body exercises. The upper body can be trained more than two times per week whereas the lower body should be trained two times. Women with intermediate experiences in RT and advanced performance level show more rapid increases in strength in the lower-body compared to the upper-body while no differences were found between upper and lower limb adaptations in RT-beginner subjects.

## 1. Introduction

Dynamic muscle strength is determined by an individual's one-repetition-maximum (1RM), which is the highest load that can be lifted once during a strength exercise with correct technique [1]. Therefore, the 1RM is also commonly used to determine individual maximal muscle strength in sport and exercise science studies, as well as in the course of an athlete's training development. It also allows the assessment of muscle development and possible imbalances in strength development in resistance training (RT) [1]. Similar to endurance training, RT with weights also brings health benefits, especially for metabolism [2]. For example, resistance training can lower fasting insulin levels and decrease insulin resistance, as well as lower systolic and diastolic blood pressure [3]. Thus, low-to-moderate intensity resistance training may also prevent arterial stiffness [4]. In addition to these general health improvements, RT has some gender-specific benefits. Studies of RT in women demonstrate that exercise-induced dilatation of the femoral arteries was greater in women than in men during leg training [5]. The World Health Organization (WHO) has recently implemented RT in their 2020 guidelines [6]. This highlights the necessity for an augmented analysis of RT-related training outcomes in women.

Unfortunately, the majority of research on RT/exercise science has been conducted with male participants. In a meta-analysis Costello and colleagues [7] examined the gender distribution of participants in more than 1,300 publications from the greater sports science field. Female subjects accounted for only 39% of participants. This current review will focus on RT with female subjects, as RT is rising in popularity as a training method, especially among young women. The question remains, however, how women can most effectively train to increase maximal dynamic muscle strength (i.e., 1RM).

One reason for different acute responses between men and women is a difference in sex hormones, which are responsible for anabolic effects after RT. Female sex hormones like estradiol possess an anabolic function due to their protein-building function in the ovaries [8]. Studies on the menstrual cycle suggest that, during the follicular phase (i.e., when estrogen concentrations are high) athletic performance and maximum strength are increased more than during the luteal phase [9–11]. Thompson and colleagues [12] postulate that a high estrogen concentration in women leads to higher release of growth hormone after RT. Two other publications recommend adapting RT to the individual phases of the menstrual cycle [13, 14]. Hagstrom and colleagues [15] recently published a systematic review with a meta-analysis on RT in young women, in which the authors found differences in the responses between the lower and upper body. The research team concluded that a strength increase of about 25% can be achieved in the upper and lower-body with a 15-week training protocol. The authors state that, for lower and upper-body training in particular, the volume and frequency of training play a determining role in increasing muscle strength.

In contrast to the review by Hagstrom and colleagues [15], the present review aims to provide practical recommendations for training protocols (i.e., repetitions per set and training

frequency per week) for the lower and upper-body RT. Only data from women who did not suffer from any known risks of hormone problems at the beginning of the included studies were used for this meta-analysis.

## 2. Methods

### 2.1 Search strategy and data sources

The PubMed database was searched in October 2021 with the following search string: 'strength training' OR 'resistance training' AND 'female' OR 'women'. Also, a filter was set to Randomized Controlled Trial (RCT) in German or English. A total of 69,383 search results were displayed. Of these, the top 10,000 matches (best matches) were downloaded on 10/24/2021. The Web of Science database was also used with the same search string on 02/18/2022. The 1,000 best matches were also downloaded here with a filter for RCTs.

Titles and abstracts of the studies were individually evaluated by two reviewers (RJ and CZ) to assess their eligibility. Any discrepancies were resolved by a third reviewer (EI). If it could already be determined in the title that a publication could be excluded due to the inclusion and exclusion criteria, which was often the case, the abstract was not checked further.

The authors of the studies that were potentially eligible were contacted for any missing data or clarification on the data presented. This review is based on the recommendations of PRISMA (Preferred Reporting Items for Systematic Reviews and Meta-Analyses).

### 2.2. Inclusion and exclusion criteria

The following inclusion and exclusion criteria were defined: (1) RCT in English or German; (2) Women in study data; (3) If both sexes were involved, female subjects' data are shown separately; (4) Subjects are reported to be mentally and physically healthy; (5) Women are older than 18 and before their expected menopause (assessed by age); (6) Measurement of dynamic strength gains before and after intervention (1RM); (7) Dynamic strength training as an intervention; (8) Duration of the training intervention at least four weeks.

Studies were included in which participants performed dynamic exercises during training. This included machine exercises as well as free weight or barbell exercises, since RT programs in commercial gyms tend to combine these techniques and methods.

Studies were excluded if it was evident that the subjects were not in good health at the start of the study. One focus here was on diseases such as obesity, being chronically underweight, extreme caloric deficits, or pre-existing mental illnesses, as these diseases carry a risk of influencing the hormonal cycle negatively [16–19].

In addition, it was not mandatory that the menstrual cycle be monitored in any way during the training period. This was not an exclusion criterion.

### 2.3. Methods of study selection

Both reviewers (RJ and CZ) used the same data file and worked independently to screen for the inclusion and exclusion criteria.

A total of 95 studies were included after the first screening using PubMed data. After searching Web of Science, an additional 23 studies were included. Of the 118 studies, a further 87 could be excluded after a full-text screening was performed. The training exercises were analysed separately, but were immediately divided into upper and lower body exercises. The aim here was to analyse the potential for eumenorrheic women to increase muscle strength gains per week divided into upper and lower-body. It was not considered relevant whether or not a workout was performed to muscular failure.

## 2.4 Statistical analysis and level of the participant's

For each of the 31 included studies, the risk of bias (Risk of Bias 2 (RoB 2)) was assessed using Cochrane guidelines. For this purpose, five criteria were specified, which were assessed as either low, unclear, or high risk. For the performance bias, a high risk was also specified if the participants and/or investigators knew, for example, whether the participants belonged to the group that trains a muscle once weekly or twice weekly, if this was the aim of the study. From the authors' point of view, this can lead to a bias, if it is to be proven that a higher frequency makes more sense. The studies were also assessed with regard to the PEDro scale.

Review Manager 5.4 and GraphPad Prism were used for the analysis and figures. The calculations for the weekly percentage increases were performed with MS Excel®. All percentage increases were divided by the duration of the intervention to determine weekly increases. The data from the meta-analyses were based solely on a comparison of weekly percentage increases. To calculate the SD in the Forest Plots, the SD of the final measurement was set in relation to the 1RM. The SD value was then divided by the duration of the studies.

In order to classify the level of participants of the included studies, the strength classification of Santos Junior and colleagues [20] was adopted. RT experienced subjects are therefore subjects above the beginner strength level. Since Santos Junior and colleagues [20] only classify strength levels for barbell exercises, the authors' specifications were used for the studies who did not include at least one barbell exercise.

## 3. Results

### 3.1. Study selection and characteristics

A total of 11,000 studies were reviewed. Of these, 31 publications met the inclusion criteria. Fig 1 shows the screening process for all 31 included studies according to PRISMA-Guidelines (Preferred Reporting Items for Systematic Reviews and Meta-Analyses).

In total, data from 621 subjects with an average age of 23.2±2.5 years could be obtained. However, in four of these studies subject age was not included. Of the 31 studies, 18 studies were conducted with beginner-level participants and 13 studies with RT-experienced women.

For the analyses the training exercises from the 31 studies were divided into upper and lower-body exercises. Lower-body training was defined as any exercise training muscles at or below the hip, while upper-body training included all exercises above hip level. The following exercises were defined as lower-body exercises. The number shows the quantity of data records of the exercises which were used for the analyses: leg press (n = 13), leg extension (n = 11), squat (n = 6), leg curl (n = 5), deadlift (n = 2), v-squat (n = 2), calf raise (n = 1), abduction (n = 1), adduction (n = 1), kick-back (n = 1), hack squat (n = 1). The following exercises were defined as upper-body exercises: bench press (n = 15), biceps curls (n = 6), chest press (n = 6), latissimus pulldown (n = 6), triceps extension (n = 4), shoulder press (n = 4), cable row (n = 2), shoulder press (n = 2), torso arm (n = 1), neck pull (n = 1), hyperextension (n = 1) and abdominal crunches (n = 1).

### 3.2. Quality of the studies and risk of bias

The risk of bias (RoB2) analysis is attached in the supporting information. It can be generally stated that there is a low risk of bias in all included studies. The selected studies were also assessed according to the PEDro methodological quality scale by Moseley and colleagues [21]. The results are presented in Table 1.

Table 1 shows the rating according to the PEDro scale. The first column lists the authors. #1 notes whether this study met the inclusion and exclusion criteria. #2 indicates whether the

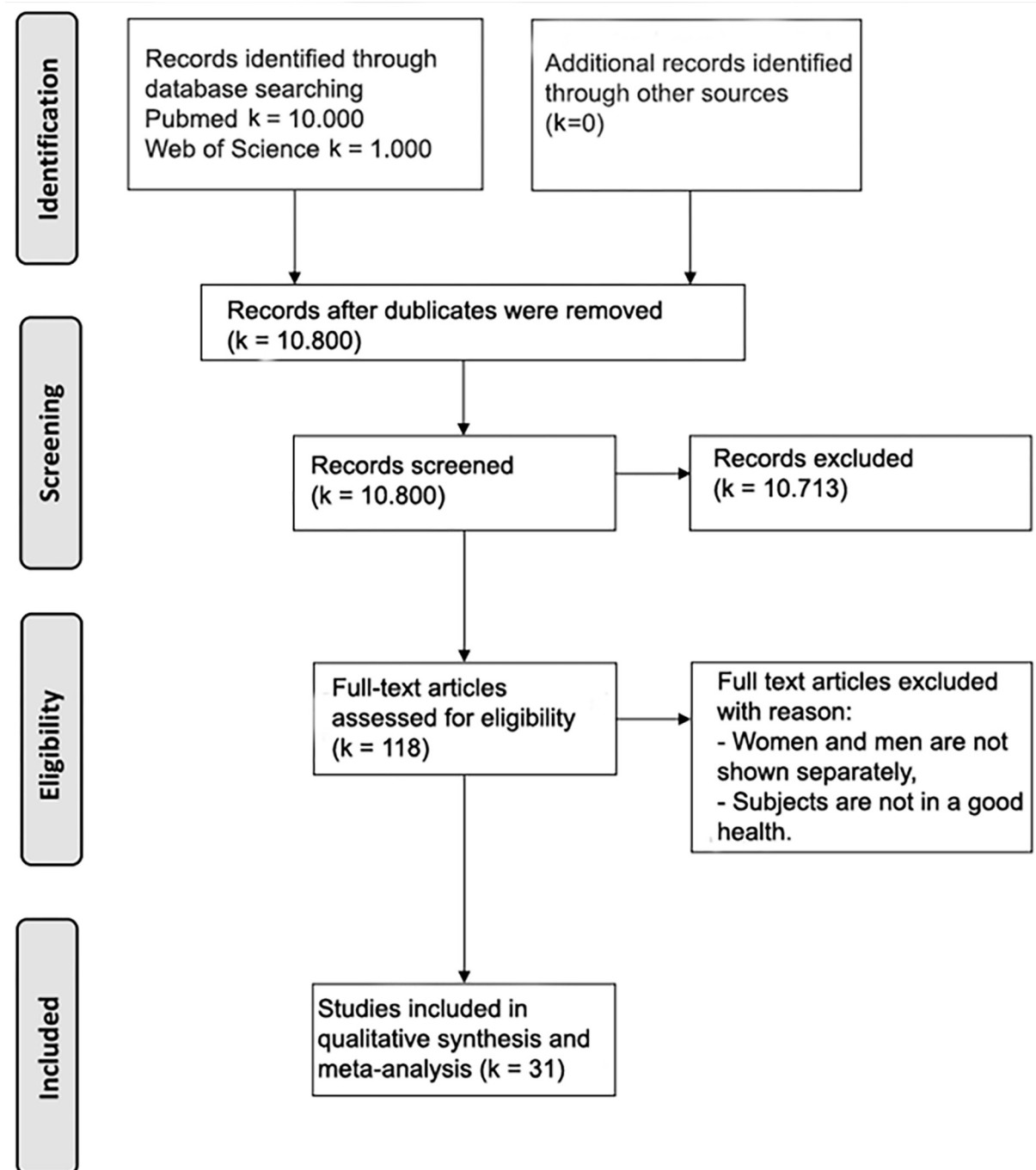

**Fig 1. PRISMA flowchart.** Review and selection process of all data records from the two databases: PubMed and Web of Science from the first data export to the final qualitative analysis.

study features randomisation of the intervention or control groups. #3 stands for a hidden allocation to the groups. #4 represents whether or not all uniform parameters were similar in all groups at the beginning. #5 stands for the blinding of participating subjects. #6 notes whether or not trainers or therapists were blinded. #7 indicates the blinding of an investigator who measured an outcome. #8 means that at least 85% of the subjects completed the study

**Table 1. The PEDro scale rating for all included studies.**

| Author | #1 | #2 | #3 | #4 | #5 | #6 | #7 | #8 | #9 | #10 | #11 | Total |
|---|---|---|---|---|---|---|---|---|---|---|---|---|
| Burt et al. [22] | 1 | 1 | 1 | 1 | 0 | 0 | 0 | 1 | 1 | 1 | 1 | 8 |
| Stefanaki et al. [23] | 1 | 1 | 1 | 0 | 0 | 0 | 0 | 0 | 1 | 0 | 1 | 5 |
| Keeler et al. [24] | 1 | 1 | 1 | 1 | 0 | 0 | 0 | 1 | 1 | 1 | 1 | 8 |
| Bell et al. [25] | 1 | 1 | 1 | 1 | 0 | 0 | 0 | 0 | 1 | 1 | 1 | 7 |
| Cacchio et al. [26] | 1 | 1 | 1 | 1 | 0 | 0 | 0 | 1 | 1 | 1 | 1 | 8 |
| Weiss et al. [27] | 1 | 1 | 1 | 0 | 0 | 0 | 0 | 1 | 1 | 1 | 1 | 7 |
| Snow-Harter et al. [28] | 1 | 1 | 1 | 1 | 0 | 0 | 0 | 1 | 1 | 1 | 1 | 8 |
| Kim et al. [29] | 1 | 1 | 1 | 1 | 1 | 0 | 0 | 1 | 1 | 1 | 1 | 9 |
| Stock et al. [30] | 1 | 1 | 1 | 1 | 0 | 0 | 1 | 1 | 1 | 1 | 1 | 9 |
| Gentil et al. [31] | 1 | 1 | 1 | 1 | 0 | 0 | 0 | 1 | 1 | 1 | 1 | 8 |
| Mayhew et al. [32] | 1 | 1 | 1 | 1 | 0 | 0 | 0 | 1 | 1 | 1 | 1 | 8 |
| Santos et al. [33] | 1 | 1 | 1 | 1 | 0 | 0 | 0 | 1 | 1 | 1 | 1 | 8 |
| de Lima et al. [34] | 1 | 1 | 1 | 0 | 0 | 0 | 0 | 0 | 1 | 1 | 1 | 7 |
| Dinyer et al. [35] | 1 | 1 | 1 | 1 | 0 | 0 | 0 | 1 | 1 | 1 | 1 | 8 |
| Marx et al. [36] | 1 | 1 | 1 | 1 | 0 | 0 | 0 | 1 | 1 | 0 | 1 | 7 |
| Mosti et al. [37] | 1 | 1 | 1 | 0 | 0 | 0 | 0 | 1 | 1 | 1 | 1 | 7 |
| Botton et al. [38] | 1 | 1 | 1 | 1 | 0 | 0 | 0 | 1 | 1 | 1 | 1 | 8 |
| Moghadasi et al. [39] | 1 | 1 | 1 | 1 | 0 | 0 | 0 | 1 | 1 | 1 | 1 | 8 |
| de Castro Cesar et al. [40] | 1 | 1 | 1 | 1 | 0 | 0 | 0 | 1 | 1 | 1 | 1 | 8 |
| Davitt et al. [41] | 1 | 1 | 1 | 1 | 0 | 0 | 0 | 0 | 1 | 1 | 1 | 7 |
| Kell [42] | 1 | 1 | 1 | 0 | 0 | 0 | 0 | 0 | 1 | 1 | 1 | 6 |
| Hostler et al. [43] | 1 | 1 | 1 | 1 | 0 | 0 | 1 | 0 | 0 | 1 | 1 | 7 |
| Schlumberger et al. [44] | 1 | 1 | 1 | 1 | 0 | 0 | 1 | 1 | 1 | 1 | 1 | 9 |
| Silva et al. [45] | 1 | 1 | 1 | 1 | 0 | 0 | 0 | 1 | 1 | 1 | 1 | 8 |
| Garcia et al. [46] | 1 | 1 | 1 | 1 | 0 | 0 | 0 | 1 | 1 | 1 | 1 | 8 |
| Burnham et al. [47] | 1 | 1 | 1 | 0 | 0 | 0 | 0 | 1 | 1 | 1 | 1 | 7 |
| Stien et al. [48] | 1 | 1 | 1 | 0 | 0 | 0 | 0 | 1 | 1 | 1 | 1 | 79 |
| Cholewa et al. [49] | 1 | 1 | 1 | 1 | 1 | 1 | 1 | 1 | 1 | 1 | 1 | 11 |
| Monteiro et al. [50] | 1 | 1 | 1 | 1 | 1 | 1 | 1 | 1 | 1 | 0 | 1 | 10 |
| Hendrickson et al. [51] | 1 | 1 | 1 | 1 | 0 | 0 | 0 | 1 | 1 | 1 | 1 | 8 |
| Rana et al. [52] | 1 | 1 | 1 | 1 | 0 | 0 | 0 | 1 | 1 | 1 | 1 | 8 |

successfully. #9 notes whether all included subjects also received any treatment or at least a control application. #10 means that at least one key outcome was statistically assessed at the end. #11 indicates that the study reported at least one point measure and at least one dispersion measure for a key outcome.

If one of the eleven attributes was fulfilled from the point of view of the evaluators, a 1 was given; if not, a 0 was set. The last column of Table 1 shows the total number of points.

One study achieved a score of 11/11. The most frequent score was 8/11, with a total of 14 studies. Only 2 studies scored less than 7/11, with only the study by Stefanaki and colleagues [23] scoring less than half.

## 3.3. Summary of all included studies

Table 2 summarizes the results of the 31 included studies. If no evaluation of the performance level could be performed according to Santos Junior and colleagues [20], the authors' data were still added. In summary, women were able to increase their 1RM in the upper body by

**Table 2. Summary of all included studies for the final analysis.**

| Author (Year) | Participants | Level of Participant's | Training protocol | 1RM exercises | Duration | 1RM increase percentage per week (summarized by upper and lower-body) | Quality (Items) |
|---|---|---|---|---|---|---|---|
| Burt et al. (2007) [22] | N: 21 Age: Colleged Age Height: 170.1 ±6.3 cm Weight: 64.3±6.9 kg | By authors: Untrained lifters. colleged age healthy women | Group 1 Per week: 1 Sets: 1 Reps: 6–10 Group 2 Per week: 2 Sets: 1 Reps: 6–10, Rest: 3 Minutes | Lower Body: Leg press | 8 weeks | Lower Body Group 1 = 37.7% Group 2 = 59.9% | 81.8% (9/11) |
| Stefanaki et al. (2018) [23] | N:13 Age: 29.7±4.7 years Height: Weight: 64.7±12.2 kg | By authors: Not engaging in more than 2 hours per week of moderate/high intensity aerobic exercise or any resistance training. healhty | Group 30% 1RM Per week: 2 Sets: 1 Reps: 30% Group 80% 1RM Per week: 2 Sets: 1 Reps: 80% | Lower Body: Leg extension Upper Body: Bicep curl | 6 weeks | Upper Body 30% = 15.4% 80% = 18.3% Lower Body 30% = 25.3% 80% = 27.2% | 90.9% (10/11) |
| Keeler et al. (2001) [24] | N: 14 Age: 32.8±8.9 years Height: 161.7 ±7.6 cm Weight: 67.9±11.5 kg | By authors: Beginner, 8 months without weighttraining | Superslow Per week: 3 Sets: 1 Reps: 50% Traditional Per week: 3 Sets: 1 Reps: 80% | Lower Body: Leg press. Leg curl. Leg extension Upp Upper Body: Torso arm. Bench press (machine). Compound row. Triceps extension. Bicep curl | 10 weeks | Upper Body Superslow (n = 6) = 1.6% Traditional (n = 8) = 3.2% Lower Body Superslow (n = 6) = 1.3% Traditional (n = 8) = 4.0% | 81.82% (9/11) |
| Bell et al. (2000) [25] | N: 9 Age: 22.3±3.3 years Height: 176 ±9.3 cm Weight: 73.4 ±11.6 kg | By authors: Experienced weightlifters, but with no training at the beginning | Per week: 3 Sets: unclear Reps: 2 till 12 | Lower Body: Leg press. leg extension | 12 weeks | Lower Body (n = 4) 63.6% | 63.6% (7/11) |
| Cacchio et al. (2006) [26] | N: 20 Age: 24.8±1 years Height: 167.4 ±4.8 cm Weight: 56.5±4 | By authors: sedentary Beginners | Per week: 3 Sets: 3 Reps: 10 (maybe not into failure) | Upper Body: Freemotion Chest Press. traditional Chest Press | 8 weeks | Upper Body FM (n = 10) = 143.6% CM (n = 10) = 71.9% | 81.8% (9/11) |
| Weiss et al. (1988) [27] | N: 28 Age: 18 to 26 (not seperatly for females) | By authors: 3 months without any training programm. healhty by questionnaire. feamles without contraceptives in the last 3 months | Per week: 3 Sets: 4 Reps: 9–13— Rest: 2 till 3 minutes | Lower Body: Seated calf raises | 8 weeks | Lower Body (n = 14) 15.3% | 72.7% (8/11) |
| Snow-Harter et al. (1992) [28] | N: 52 Age: 19.9±0.7 years Height: 165±7.3 cm Weight: 60.4±12.8 kg (also with data from the runners group) | By Santos Junior et al. [19]: Intermediate Bench Press (57%) By authors: No competitive athletes. min. 8–12 menstrual cycles in a year for the last 3 years. | Per week: 3 Sets: 3 Reps: 65–85%. at the beginning less then 65% | Upper body: Bicepscurl. Triceps extension. Militarypress. Facepulls. Benchpress. Back extension. lat pulldown Lower Body: Leg extension. Leg curl. Abduction. | 8 months | Upper Body (n = 12) = 23.7% Lower Body (n = 12) = 44.1% | 81.8% (9/11) |
| Kim et al. (2011) [29] | N: 35 Age: 20.5± 0.4 years Height: 166.9±1.53 cm Weight: 66.6±5.4 kg | By authors: No strength training or aerobic endurance training for at least 6 months. | TRT Group (n = 13) Per week: 3 Sets: 3 Reps: 80% SRT Group (n = 14) Per week: 2 Sets: 1 Reps: 50% | Upper Body: Shoulder press. chest press. rowing and lat pulldowns Lower Body: Leg press | 4 weeks | Upper Body TRT (n = 13) = 5.5% SRT (n = 14) = 3.8% Lower Body TRT (n = 13) = 5.9% SRT (n = 14) = 3.7% | 81.8% (9/11) |

(*Continued*)

**Table 2.** (Continued)

| Author (Year) | Participants | Level of Participant's | Training protocol | 1RM exercises | Duration | 1RM increase percentage per week (summarized by upper and lower-body) | Quality (Items) |
|---|---|---|---|---|---|---|---|
| Stock et al. (2016) [30] | N: 47 Age: 21±3 years Height: 162.1 ±9.6 cm Weight: 63.3±11 kg | By Santos Junior et al. [19]: Beginner Sqauts and Deadlift By authors: No weightlifting for the past 6 months. healthy | Lower Per week: 3 Sets: maybe 4 (unclear) Reps: 5 Moderate Per week: 2 Sets: maybe 4 (unclear) Reps: 5; Rest: 3 minutes | Lower Body: Squats, Deadlifts | 4 weeks | Lower Body Lower Volume (n = 15) = 95.8% Moderate Volume (n = 16) = 86.2% | 81.8% (9/11) |
| Gentil et al. (2017) [31] | N: 8 Age: 34.1±4.3 years Height: 166±0.1 cm Weight: 70.1±9.3 kg (only Resistance training group) | By authors: Eumenohheic. weight stable for 6 months. inactive. be free of medical problems that could be aggravated by the study protocol. | Per week: 3 Sets: 3 Reps: 8–12 | Upper Body: Bicepcurl with barbell Lower Body: Leg extension | 8 weeks | Upper Body RT (n = 8) = 22.3% Lower Body RT (n = 8) = 34.6% | 90.9% (10/11) |
| Mayhew et al. (2011) [32] | N: 62 Age: 19.1±0.8 years Height: 164.0 ±5.6 cm Weight: 62.1±11.5 kg | By Santos Junior et al. [19]: Intermediate (49%) By authors: Healthy without any training in the last 6 months | Per week: 3 Sets: 3 Reps: 6–12 Rest: 2 Minutes | Benchpress. Squats. Lat pulldowns. Calf raises. arm curls. Shoulder press | 12 weeks | Upper Body 23.4% | 81.8% (9/11) |
| Santos et al. (2010) [33] | N: 16 Age: 25.4 ±1.95 years Height: 162.7±4.4 cm Weight: 57.7±3.9kg (only the two intervention groups AA + AST) | By authors: Sedentary. They did not perform any other physical activity during the intervention. Healhy | Per week: 2–3 (every second day) Sets: 3 Reps: 10–12 | Upper Body: Machine Benchpress | 8 weeks | Upper Body AA (n = 8) = 22% AST (n = 8) = 42.5% | 72.7% (8/11) |
| de Lima et al. (2012) [34] | N: 20 Age: 26.3 ±3.58 years Height: 163.5±0.1y Weight: 62.93±8.8cm (only the two intervention groups LP + DUP) | By Santos Junior et al. [19]: Intermediate Linear (53%) and Daily Undulating (51%). By authors: Healthy. range: 20–35 years old. non-obese. no training in 6 months | Per week: 2 Sets: 3–4 Reps: 15–30 (not very clear) Rest pause: 1–2 minutes | Upper Body: Benchpress. Bicepscurl Lower Body: Legpress | 12 weeks | Upper Body Linear (n = 10) = 19.3% daily undulating (n = 10) = 22% Lower Body Linear (n = 10) = 48.2% daily undulating (n = 10) = 38.4% | 90.9% (10/11) |
| Dinyer et al. (2019) [35] | N: 23 Age: 21.2±2.2 years Height: 167.1 ±5.7 cm Weight: 62.3±16.2 kg | By authors: 2 years without any weighttraining and less then 2 years of any sports activity | Per week: 2 Sets: 2–3 Reps: 6–11; Rest pause: 1,5 minutes | Upper Body: Shoulder press, Lat pulldown Lower Body: Leg extension, Leg curl | 12 weeks (9 weeks of training) | Upper Body 30% (n = 11) = 26.9% 80% (n = 12) = 27.4% Lower Body 30% (n = 11) = 12.9% 80% (n = 12) = 12.5% | 81.8% (9/11) |
| Author (Year) | Participants | Level of Participant's | Training protocol | 1RM exercises | Duration | 1RM increase percentage per week (summarized by upper and lower-body) | Quality (Items) |
| Marx et al. (2001) [36] | N: 34 Age: 22.7±4.6 years Height: 165.7±5.2 cm Weight:: 56.2±6.3 kg (with control group) | By Santos Junior et al. [19]: Beginner Benchpress (Single set = 39% and high volume = 37%) By authors: All subjects had a regular menstrual cycle of 28 to 32 days in the past year. No oral contraceptives were taken. | Per week: 2 Sets: 2–3 Reps: 6–11 | Upper Body: Benchpress Lower Body: Leg press | 24 Wochen | Upper Body Singe set (n = 12) = 12.2% High-volume (n = 12) = 46.8% Lower Body Single set (n = 12) = 11.2% High-volume (n = 12) = 31.8% | 90.9% (10/11) |

(*Continued*)

**Table 2.** (Continued)

| Author (Year) | Participants | Level of Participant's | Training protocol | 1RM exercises | Duration | 1RM increase percentage per week (summarized by upper and lower-body) | Quality (Items) |
|---|---|---|---|---|---|---|---|
| Mosti et al. (2014) [37] | N: 30 Age: 22.1±2.2 years Height: 168.2±7 cm Weight:: 65.3±9.3 kg (with control group) | By authors: no weighttraining of the upper body in the last 6 months | Per week: 3 Sets: 4 Reps: 3–5 | Lower Body: Hack squat with high speed | 12 weeks | Lower Body (n = 14) 83.1% | 81.8% (9/11) |
| Botton et al. (2016) [38] | N: 43 Age: 23.9±2.6 years Height: 162.3±6.2 cm Weight:: 58.6±5.6 kg (with control group) | By authors: All subjects less then 3 months without any weight training. 6 subjects take no oral contraceptives | Per week: 2 Sets: 2–3 Reps: 5–15 | Lower Body: Leg extension | 12 weeks | Lower Body UG-Group (n = 14) bilateral: 19.5% unilateral: 32.1% BG-Group (n = 15) bilateral: 27.5% Unilateral: 23.5% | 90.9% (10/11) |
| Moghadasi et al. (2011) [39] | N: 20 Age: 25.3±3.2 years Height: no data Weight: no data | By authors: Healthy, last 6 months without any weighttraining | Per week: 3 Sets: 2–4 Reps: 8–12; Rest: 2–3 minutes | Upper Body: Chest Press. Shoulder Press. Lat pulldown. Bicep curl. Triceps pulldown Lower Body: Leg press. Leg extension. Leg curl | 12 weeks | Upper Body (n = 10) 96.8% Lower Body (n = 10) 40.9% | 90.9% (10/11) |
| de Castro Cesar et al. (2019) [40] | N: 20 Age: 20.7±2.1 years Height: 1.66 ±0.1cm Weight: 57.7±8.4kg | By authors: in the last 3 months without any weighttraining. healthy | Per week: 3 Sets: 3 Reps: 15; Rest between sets: 1 minute | Upper Body: Chest press. Lat pulldown. Militarypress. Tricep extension. Bicepscurl Lower Body: Leg oress. Leg extension. Leg curl | 12 weeks | Upper Body (n = 9) = 21.7% Lower Body (n = 9) = 35.7% | 72.7% (8/11) |
| Davitt et al. (2014) [41] | N: 28 Age: 19.8±0.2 years Height: no Weight: 61±2.5 kg (weightlifters and endurance training group) | By Santos Junior et al. [19]: Intermediate Benchpress (60%) | Per week: 4 Sets: 3 Reps: 8–12; Rest: 1–1,5 minutes | Upper Body: Benchpress Lower Body: Leg press | 8 weeks | Upper Body (n = 10) = 24.3% Lower Body (n = 10) = 38.6% Only data from the group weighlifting before endurance training. | 81.8% (9/11) |
| **Author (Year)** | **Participants** | **Level of Participant's** | **Training protocol** | **1RM exercises** | **Duration** | **1RM increase percentage per week (summarized by upper and lower-body)** | **Quality (Items)** |
| Kell (2011) [42] | N: 20 Age: 22.5 ±4.6 years Height: 1. 70 ± 0.1 m Weight: 59.4 ± 5kg (only female weighttraining) | By Santos Junior et al. [19]: Intermediate Benchpress (58%) and Sqaut (82%). By authors: More then 11 months of weighttraining before the study. healthy | Per week: 4 Sets: 3 Reps: 8–12 | Upper Body: Benchpress. Latpulldown. Shoulder press barbell Lower Body: Squats | 12 weeks | Upper Body = 37% Lower Body = 43.8% | 81.8% (9/11) |
| Hostler et al. (2001) [43] | N: 10 Age: 20.9±1.1 years Height: 163.6±7.6 cm Weight:: 58.9±5.3 kg Only female subjects. | By Santos Junior et al. [19]: Advanced Benchpress (SI-Group = 63% und TRAD-Group = 60.9%). By authors: 6 months without any weighttraining. healthy | Per week: 2 Sets: 3 Reps: 60% till muscle failure, Rest: 3 minutes | Upper Body: Benchpress, Triceps. | 8 weeks | Upper Body TRAD-Group (n = 5) = 12.2% SI-Group (n = 5) = 10.3% | 81.8% (9/11) |

(*Continued*)

**Table 2.** (Continued)

| Author (Year) | Participants | Level of Participant's | Training protocol | 1RM exercises | Duration | 1RM increase percentage per week (summarized by upper and lower-body) | Quality (Items) |
|---|---|---|---|---|---|---|---|
| Schlumberger et al. (2001) [44] | N: 27 Age: 26.3±5.1 years Height: 166.6 ±5.4cm Weight:: 65.37±8.67 kg with control group | By authors: healthy, at least 6 months of weighttraining experience. | Single set: Per week: 2 Sets: 1 Reps: 6–9 Multi set: Per week: 2 Sets: 3 Reps: 6–9, Rest: 2 minutes (MS) | Upper Body: Chest press Lower Body: Leg press | 6 weeks | Upper Body MS-Group (n = 9) = 10.4% Single-set Group (n = 9)) = 4.1% Lower Body MS-Group (n = 9) = 15.8% Single-set Group (n = 9) = 4.1% | 81.8% (9/11) |
| Silva et al. (2012) [45] | N: 12 Age: 23.5±2.5 years Height: 165.8 ±6.5 cm Weight: 59.2±8.2 kg | By Santos Junior et al. [19]: Intermediate Benchpress (49.8%). By authors: healthy. 3 months without weighttraining | Per week: 2 Sets: 2–3 Reps: 8–18 | Upper Body = Benchpress Lower Body = Leg press. Leg extension | 11 weeks | Upper Body = 20% Lower Body = 45.9% | 72.7% (8/11) |
| Garcia et al. (2016) [46] | N: 11 Age: 25.2±5.3 years Height:— Weight: 59.9±4.8kg | By Santos Junior et al. [19]: Both groups: Sqaut Advanced (MS = 103%; TRI = 102.3%) and Deadlifts Intermediate (MS = 96%; TRI = 105.6%). By authors: At least 12 months of weighttraining experience. Healthy | Per week: 3 Sets: 3 Reps: 6–14 | Lower Body: Sqaut, Deadlift | 12 weeks | Lower Body MS-Group = 20.6% TRI-Group = 21.6% | 81.8% (9/11) |
| Burnham et al. (2010) [47] | N: 19 Age: 19.8±1.6 years Height: 179.8±4.7 cm Weight: 74.9±6.7 kg | By Santos Junior et al. [19]: Both groups: Bench Press Advanced (62%). By authors: At least 1 year of experience. | Per week: 2 Sets: 3 Reps: 80–90% One Group with Weight chains | Upper Body: Benchpress | 8 weeks | Upper Body Traditional (n = 9) = 11.9% Chain (n = 10) = 17.4% | 81.8% (9/11) |
| **Author (Year)** | **Participants** | **Level of Participant's** | **Training protocol** | **1RM exercises** | **Duration** | **1RM increase percentage per week (summarized by upper and lower-body)** | **Quality (Items)** |
| Stien et al. (2020) [48] | N: 38 Age: 22.26 ±1.24 years Height: 166.7±2.7 cm Weight: 66.6±5.3 kg | By authors: healthy. Physically active women. on average 9.6 months of strength training experience | Per week: 2–3 Sets: 3–4 Reps: 6–10 | Lower Body: Legpress. Leg extension, Kick Back | 8 weeks | Lower Body Single Joint Group (n = 18) = 16.7% Multi Joint Group (n = 20) = 19.8% | 72.7% (8/11) |
| Cholewa et al. (2018) [49] | N: 23 Age: 20.9±1.4 years Height: 165.6 ±6.4 cm Weight: 68.7±11.9 kg | By Santos Junior et al. [19]: Beta-Group Intermediate (Sqaut = 63.2%; Benchpress = 36.2%); Placebo-Group Intermediate (Sqaut = 85%; Benchpress = 50.2%). By authors: No weighttraining in the last 6 months. healthy | Upper Body: Per week: 1 Sets: 3 Reps: 8–12 Lower Body: Per week: 2 Sets: 3 Reps: 8–12 Rest: 2–3 minutes | Upper Body: Benchpress Lower Body: Sqaut | 10 weeks | Upper Body Beta-Group (n = 11) = 9.1% Placebo-Group (n = 12) = 14.5% Lower Body Beta-Group (n = 11) = 31.2% Placebo-Group (n = 12) = 29.2% | 81.8% (9/11) |
| Monteiro et al. (2008) [50] | N: 20 Age: 36.9 ±1.5years Height: 157.9±9.9 cm Weight: 64.9±10.5 kg | By authors: seated subjects. at least 6 months without any physical activity | Per week: 3 Sets: 3 Reps: 8–12 | Upper Body: Benchpress (without Abdominal Crunch) Lower Body: Hack machine. Smith Machine | 10 weeks | Upper Body (n = 10) = 56.1% Lower Body (n = 10) = 68.1% | 72.7% (8/11) |

*(Continued)*

**Table 2.** (Continued)

| | | | | | | |
|---|---|---|---|---|---|---|
| Hendrickson et al. (2010) [51] | N: 18 Age: 21±0.5 years Height: 164.7±1.9 cm Weight: 64.5±1.9 kg (only resitant training group) | By Santos Junior et al. [19]: Intermediate Benchpress (48%) and Sqaut (83%). By Authors: Subject are healthy and have a menstrual cycle | Per week: 3 Sets: 3 Reps: 3–12 | Upper Body: Benchpress Lower Body: Squat | 8 weeks | Upper Body (n = 17) = 22.2% Lower Body (n = 17) = 14.2% | 90.9% (10/11) |
| Rana et al. (2008) [52] | N: 34 Age: 21.1±2.7 years Height:164.78 ±5.5 cm Weight: 66.3±9.9 kg with control group | By authors: Healthy subjects | Per week: 2–3 Sets: 3 TS-Group: Reps: 6–10 TE-Group Reps: 20–30 | Lower Body: Legpress, Sqaut. Leg extension | 6 weeks | Lower Body TS-Group (n = unclear) = 54.2% LV-Group (n = unclear) = 27.9% TE-Group (n = unclear) = 20.9% C-Group (n = unclear) = 3.2% | 81.8% (9/11) |

7.2% and in the lower body by 5.2% per week. The quality of every study is also listed in Table 2 (last column).

**3.4.1 Upper- and lower-body comparison.** Of the 31 included studies, only 13 studies trained with an identical load (repetitions per set, sets per workout, frequency per week) with at least one lower-body and one upper-body exercise.

Only one study [51] found that the upper body strength increases more rapidly than the lower body in intermediate and advanced subjects. The authors reported the participants as moderately active in recreational activities with fewer than two training sessions per week. According to Santos Junior and colleagues [20] strength levels on the bench press and squat were rated as intermediate. Fig 2 shows the Forest Plot for RT experienced subjects.

For beginners, the study by Moghadasi and colleagues [39] greatly affected the results. Triceps extension increased by 260% in twelve weeks. There was also a 130% increase in the 1RM in chest press. If this study would be removed from the results the mean difference would change from 0.47 to 0.77 and the 95% CI would change from 0.22 to 1.31 in favor for lower body. Fig 3 shows the Forrest Plot for subjects with beginner level.

When compared to beginner subjects RT-experienced subjects enjoyed a preferable overall effect and more rapid lower-body strength increases. Overall, for Fig 2 (experienced subjects) and 3 (beginner subjects), the pooled effect size is 0.47 with a 95% CI of -0.13 and 1.08.

Weekly muscle strength gains were 3.7% higher in the lower body than in the upper body. For intermediate and advanced subjects the gains were an impressive 56.7% higher.

**3.4.2. Repetitions and frequency per week for upper and lower-body strength.** For the analysis of repetitions per set, the following division was used: i) 1 to 6 repetitions, ii) 6 to 13

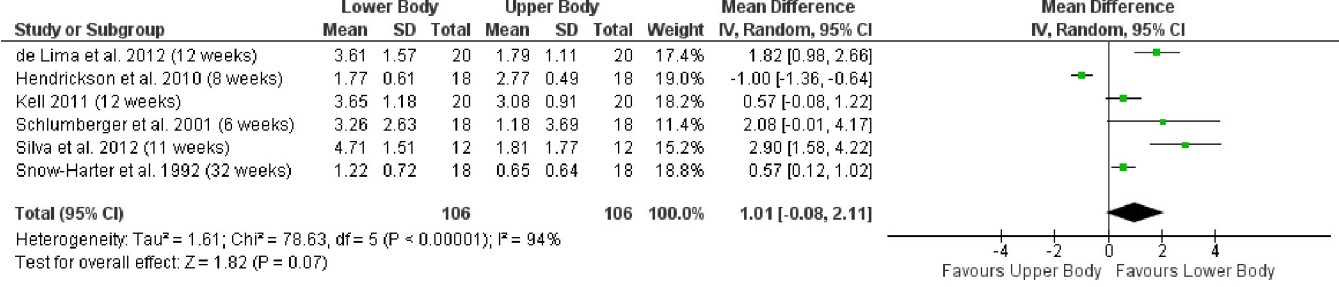

**Fig 2. Forrest plot.** Lower-body gains compared to upper-body 1RM gains in intermediate and advanced subjects.

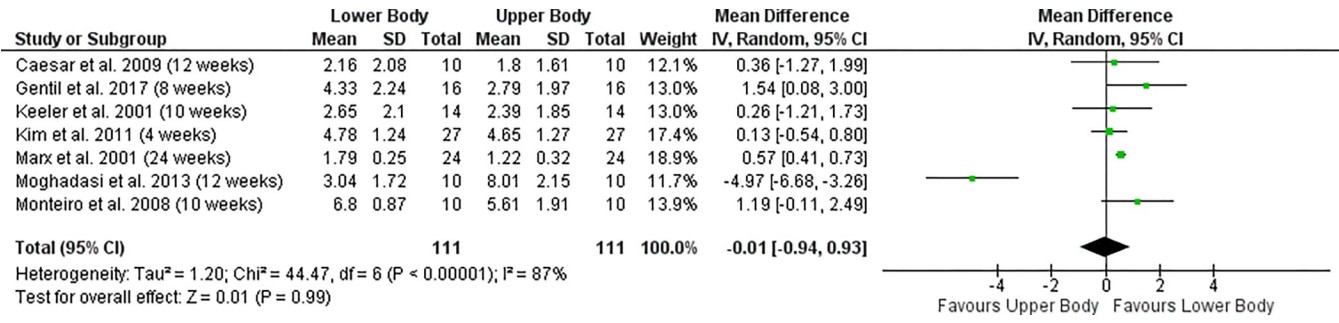

**Fig 3. Forrest plot.** Lower-Body gains compared to upper-body 1RM gains in beginner subjects.

repetitions, iii) 13 to 20 repetitions, and iv) more than 20 repetitions to ensure a better overview. Of the 31 studies included a total of 29 data sets for the lower body and 25 data sets for the upper body were obtained. Often, different training variables were tested within the studies, such as low-load vs. high-load RT. Therefore, a study can include up to four records on 1RM gains in our final analysis. Some studies, however, only provided one set of records, such as frequency per week but or number of repetitions per set. Fig 4 shows the results of this analysis.

**3.4.3. Information on menstrual status in the included studies.** Of the 31 included studies, only four gave information about the menstrual cycle of their subjects. These data are very difficult to evaluate, as two studies by Hendrickson and colleagues [51] and Gentil and colleagues [31] merely state that participants were eumenorrheic at the beginning of the intervention. Only two studies [27, 38] report the use of contraceptives among participants. Therefore, this quality item was also awarded if menstrual status was checked in any way at the beginning of the study. No study examined hormonal levels or function, nor did they track menstrual cycle health during the study. Details are shown in Table 2.

## 4. Discussion

Key Points:

- Weekly percentage 1RM gains are higher in the lower body than in upper body in RT-experienced women.

- Training the lower-body with a maximum of six repetitions per set and the upper-body with a repetition range of 13 to 20 per set seems to be most effective for increases in 1RM.

- The ideal training frequency for the lower body is twice weekly, while for the upper body, two to three times per week results in the highest 1RM increases.

- Women with intermediate experiences in RT and advanced performance level show more rapid increases in strength in the lower-body compared to the upper-body while no differences were found between upper and lower limb adaptations in RT-beginner subjects.

This review provides specific recommendations on the number of repetitions per set and weekly training frequency for dynamic strength increases in eumenorrhoeic women. The quality of the included studies was considered good overall, as far as this can be proven, by a risk of bias analysis and PEDro scale (average scores: 7.8 points).

A total of 621 subjects from 31 included studies were able to increase their upper-body strength by 7.2% and lower-body strength by 5.2% per week respectively. These values are independent of the weights, sets, exercises, and training status of the subjects. Fig 4 shows the

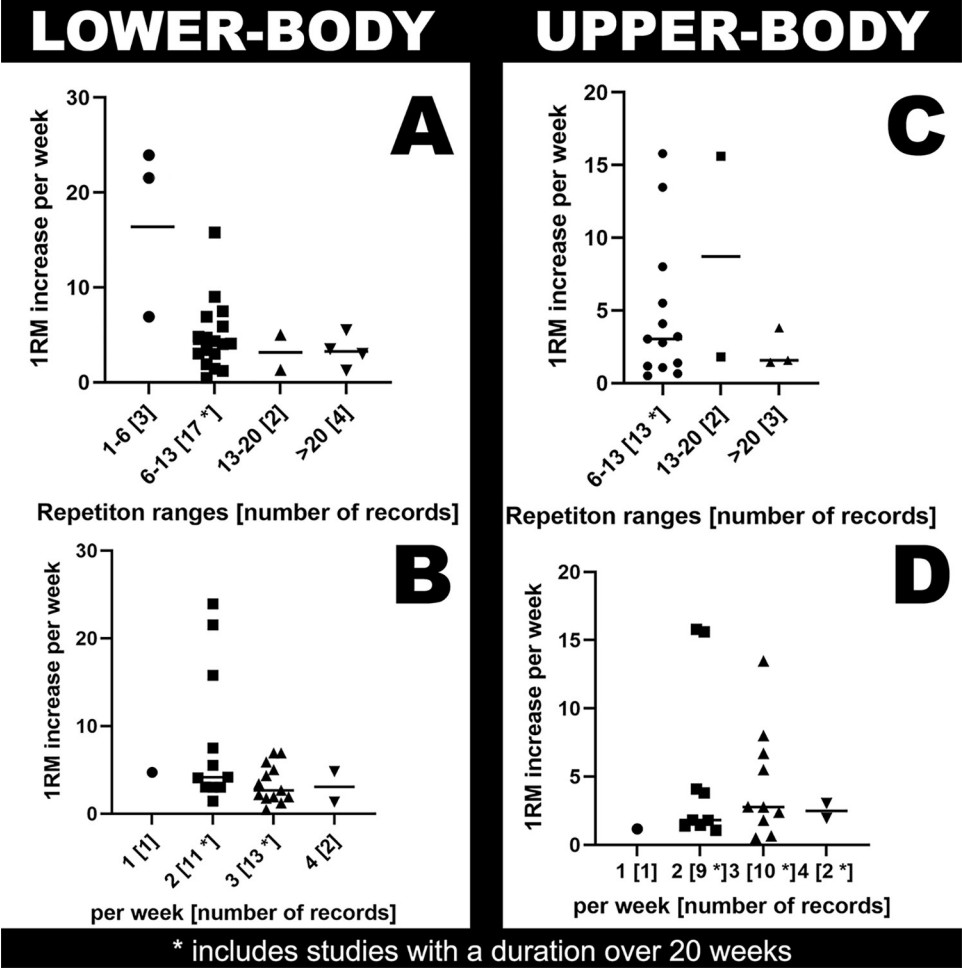

**Fig 4. Evaluation for repetitions per set and training frequency per week for all included studies.** (A) Lower-body repetitions per set, (B) lower-body training frequency per week, (C) upper-body repetitions per set and (D) upper-body training frequency per week.

evaluation for repetitions per set and training frequency per week. Here training recommendations for effectively increase the 1RM can be drawn. Upper-body exercises should be performed with more repetitions per set (13 to 20 repetitions) to achieve the highest dynamic strength gains compared to lower-body exercises. The lower body should be trained with heavy loads and thus fewer repetitions (1 to 6 repetitions per set). One reason for this could be the differences in the distribution of muscle fibers. Most muscle groups in the upper body contain a larger proportion of fast-twitch fibers than those in the lower body, so adaptation may vary with in muscles in the upper-body and in the legs [53, 54]. Muscles in the upper body, excluding abdominal and lower-back muscles, are used for powerful movements in everyday life and may respond better to completely new stress with lower weights and higher repetitions. For example, two studies [28, 50] deliver only a single record for upper-body strength in our final analysis. We did not analyze if there are differences between exercises which are mainly focusing on the legs compared to lower body exercises which involve more muscles than just leg muscles (e.g., back muscles in the deadlift).

The muscles of the upper body are mainly used for powerful everyday tasks. Here, the best gains were found with 13 to a maximum of 20 repetitions per set. However, hip and leg

musculature respond better to a powerful, intense load with high weights and few repetitions (1–6 repetitions per set). This could have been an advantage, especially in studies with beginners, as a completely new stimulus to the muscle. Muscles in the upper body most often have a smaller volume than muscles in the lower body [55]. As previously mentioned, muscles in the upper body are under less stress in everyday life. Moreover, women have a higher percentage of type 1 muscle fibers than men, which may contribute to a quicker recovery, so that women may, in general, be able to exercise one muscle more frequently than men [56, 57].

It is possible than the cause of varying adaptations in the upper and lower body in women is also due to the menstrual cycle phases. A recent paper by Kissow and colleagues [58] found that the muscle hypertrophy in the legs is higher in the follicular phase, when only estrogen is high. In the upper body, however, no differences in adaptation during different hormonal phases could be detected. Accordingly, it can be assumed that the legs react more sensitively to the different hormone phases [58]. There appears to be variance in muscular adaptations in young women between the upper and lower body as well. Estrogen mobilizes growth hormones more than progesterone and counteracts muscle protein breakdown after training [58]. Perhaps this is more relevant for muscle groups with larger muscle mass, such as the quadriceps. This seemed especially the case in comparisons of training a squat and a bench press, which was a common protocol in the included studies of this review. In men, it appears that more sex hormones are produced during and immediately after a squat than during a bench press [59]. This is where further research is important to determine the adaptations between the upper and lower body and hormonal reactions prior to RT in women specifically in order to make further training recommendations. The course and effects of the individual menstrual cycle phases should also be recorded.

Fig 2 (Forest Plot) shows that when RT-experienced women trained with an equal load for the upper and lower body, the percentage weekly gains were higher for lower-body exercises. Only one study by Hendrickson and colleagues [51] achieved better strength gains in the upper body than in the lower body. Apparently, significant increases were achieved here with push exercises for chest and triceps. Nevertheless, it is unclear why the upper body is so clearly ahead of the lower body in this study.

The systematic review by Hagstrom and colleagues [15] also found preferable strength gains in the lower body compared to the upper body (around 2%). Here the findings from this review are in line with the results of Hagstrom and colleagues [15]. It can be speculated that young female subjects are more interested in increasing their performance on a leg press than on a chest press. It is possible that female participants are more familiar with leg exercises in general and therefore leg press, knee extensions and squats are performed more often in their daily training compared to lower-body exercises. For the upper body, bench press, biceps curl, and latissimus pull-downs were performed. Overall, these six exercises accounted for over 60% of all exercises in the included studies. Some of the included studies stated that subjects were verbally encouraged to train into muscle failure. This is questionable, as half of the studies were conducted with beginners, who are unlikely to have much experience of when muscle failure occurs and how it feels.

However, it should again be noted that none of the studies tested the hormonal status of the subjects during the course of the study. Also, strength measurements were never related or compared to the current hormone cycle phase. Therefore, the selected literature cannot provide precise information on how individual menstrual cycle phases and, thus, the female sex hormones estrogen and progesterone affect training success. Despite strict guidelines, it cannot be excluded that the included studies also involve results from women who have an irregular menstrual cycle or hormone profile. In addition, it should be noted that Jansen de Jonge et al. [16] already pointed out that a lack of control of the diet may lead to hormonal problems.

Dietary intake and female sex hormones are, therefore, of high importance in relation to female RT and should be investigated further.

Given that almost no study monitored diet of the subjects, such as caloric intake or macro-nutrients, the authors' data are difficult to evaluate. Daily nutrition has a significant impact on regeneration and adaptation after sports in general, especially after RT [60]. Not only are proteins important to consume, but also a certain amount of fat. For example, in a study by Trexler and colleagues [61], subjects that consumed more than 35% of fat in their total calories and significantly increased their bench press performance in 1RM compared to the group that consumed less than 35% daily. Additionally, participants who consumed more fat lost more body fat at the end of the study compared to the participants who consumed less fat [61]. Luteinizing hormone (LH) levels were measured in a study of 29 regularly menstruating women with controlled exercise and energy intake. The release of the LH hormone was impaired in most subjects when the daily calorie intake was restricted to 30 kcal per lean body mass [62]. As previously mentioned, it is important that young female subjects have a cyclical increase in LH in the late follicular phase to make sure that ovulation takes place [16].

## 5. Limitations

Half of the included studies were conducted with beginners to RT, often defined as "untrained" by the authors of the studies. As such, it can be assumed that they did not have much experience with an appropriate diet to support RT and could potentially been under- or over-fueled. In addition, in Fig 4 the data from supposed beginners is combined with data from advanced, RT-experienced subjects, which could also distort the outcomes. It is of course possible that, as stated by Jansen de Jonge and colleagues [15], data could be included from women who suffered from hormonal imbalances or irregular menstruation. Such issues can arise as a result of chronically intense physical activity, poor recovery, and a lack of diet monitoring. Since, the purpose of this review is to examine the effects of RT on young women, this is certainly the greatest limitation.

This review combines studies that use different training techniques like circuit training, volume training, and even training with slow or fast reps. Some studies trained into muscle failure, whereas some did not, which may have influenced the results as well. Other training variables, such as time under tension (TUT) or range of motion (ROM), were not included in our analysis for upper and lower-body strength gains. Worthy of note is that our meta-analysis includes women with and without hormonal contraceptives. There is evidence that rest time can have an influence on hormone responses during RT [63].

## 6. Conclusion

Young healthy women can increase their maximum muscle strength in the lower-body by 7.2% and by 5.2% in the upper-body per week through resistance training. Based on our results it seems that the upper-body can be trained with lighter weights and more repetitions to increase the 1RM compared to the lower body, which can be trained with up to six repetitions per set. The lower-body should be trained two times per week for optimal 1RM gains, while training two to three times a week resulted in highest strength gains in the upper body. This includes exercises with a barbell, machines, or a cable. In the studies where at least one upper and one lower-body exercise was performed with the same training load (repetitions per set, sets per workout, training frequency per week), more lower-body than upper-body strength was built. Interestingly, women with intermediate experiences in RT and advanced performance level show more rapid increases in strength in the lower-body compared to the upper-body while no differences were found between upper and lower limb adaptations in RT-

beginner subjects. Future studies on female participants should focus more closely on the variance in adaptations in the lower and upper body, since other authors have already found differences here as well. Future studies should also closely control the hormonal phases in their female subjects and include dietary guidance or monitoring in their analysis.

## Supporting information

**S1 Checklist. PRISMA 2009 checklist.**
(PDF)

**S1 Appendix. RoB-analysis.**
(TIF)

## Author Contributions

**Conceptualization:** Roger Jung, Sebastian Gehlert.

**Formal analysis:** Roger Jung, Eduard Isenmann.

**Investigation:** Roger Jung.

**Project administration:** Roger Jung, Christoph Zinner.

**Resources:** Stephan Geisler, Eduard Isenmann.

**Supervision:** Sebastian Gehlert, Stephan Geisler.

**Validation:** Eduard Isenmann.

**Visualization:** Roger Jung, Christoph Zinner.

**Writing – original draft:** Roger Jung, Christoph Zinner.

**Writing – review & editing:** Julia Eyre, Christoph Zinner.

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
