## [Decision Letter · Decision Letter 0]

16 Jan 2023

PONE-D-22-24644Weekly percent 1RM gains are higher in the lower-body than the upper-body in resistance training experienced healthy young women - a systematic review with meta-analysis about female resistance trainingPLOS ONE

Dear Dr. Jung,

Thank you for submitting your manuscript to PLOS ONE.  We have now received three reviews for your manuscript. While all three reviewers found your work interesting and of high standard, they expressed numerous concerns regarding various issues regarding content and details. Therefore, after careful consideration, we feel that your manuscript has merit but does not fully meet PLOS ONE’s publication criteria as it currently stands. We invite you to submit a revised version of the manuscript that addresses all concerns raised by the reviewers. We look forward to your revised manuscript.

We look forward to receiving your revised manuscript.

Kind regards,

Hans-Peter Kubis, PD. Dr. rer. nat.

Academic Editor

PLOS ONE

and https://journals.plos.org/plosone/s/file?id=ba62/PLOSOne_formatting_sample_title_authors_affiliations.pdf.

“NO”

“NO authors have competing interests”

Reviewers' comments:

Reviewer's Responses to Questions

**Comments to the Author**

1. Is the manuscript technically sound, and do the data support the conclusions?

Reviewer #1: Partly

Reviewer #2: Yes

Reviewer #3: Partly

2. Has the statistical analysis been performed appropriately and rigorously? 

Reviewer #1: Yes

Reviewer #2: Yes

Reviewer #3: Yes

3. Have the authors made all data underlying the findings in their manuscript fully available?

Reviewer #1: Yes

Reviewer #2: Yes

Reviewer #3: Yes

4. Is the manuscript presented in an intelligible fashion and written in standard English?

Reviewer #1: Yes

Reviewer #2: Yes

Reviewer #3: Yes

5. Review Comments to the Author

Reviewer #1: General comments:

The manuscript: „WEEKLY PERCENT 1RM GAINS ARE HIGHER IN THE 2 LOWER-BODY THAN THE UPPER-BODY IN RESISTANCE 3 TRAINING EXPERIENCED HEALTHY YOUNG WOMEN - A 4 SYSTEMATIC REVIEW WITH META-ANALYSIS ABOUT 5 FEMALE RESISTANCE TRAINING“, deals with real Research gap in sport science. I agree that there is a lack of women’s studies in resistance training, but the authors gathered articles by well-defined criteria.

There one issue in the conclusion statement. Authors are presenting that lower body gains in RM are higher than upper body. However, the difference 4.3% per week in the upper body and 4.7% per week in the lower-body exercises, does not seems to be relevant to this conclusion. Did you make statistics on this? What are minimal detectable changes for % of RM? (please reference this).

The introduction of the manuscript very briefly presents basic resistance training variables, where the main parameter is exercise selection. The exercise selection might be whole body training, upper/lower limb training, isolated training etc. Please define what you want to understand as lower body training. Many lower body exercise like deadlifting and squatting, includes a lot of back and trunk muscle involvement, how you can distinguish the upper and lower body training. This should be mentioned in the inclusion criteria and discussion (including muscle group size). The absence of full body training is one of the main limitations of this study, try to mention that in the discussion.

The conclusion of the study should be clarified by reporting separately on beginners and advanced resistance-trained women. This should be in the abstract as well since there are

The study discussion should mention all loading parameters of resistance training.

Therefore I recommend the article after major revisions.

Specific comments:

Line 53: the bullet point “Weekly percentage 1RM gains are higher in training experienced women in lower body rather than in upper-body exercises“ seems to be obvious to add by how much they can strengthen more in lower limbs.

Line 69: What about arterial stiffness?

Jurik, R., Żebrowska, A., & Stastny, P. (2021). Effect of an Acute Resistance Training Bout and Long-Term Resistance Training Program on Arterial Stiffness: A Systematic Review and Meta-Analysis. Journal of clinical medicine, 10(16), 3492.

In introduction or discussion, try to justify whether hormonal response of man and women really differ and how.

Hakkinen, K., Pakarinen, A., Kraemer, W. J., Newton, R. U., & Alen, M. (2000). Basal concentrations and acute responses of serum hormones and strength development during heavy resistance training in middle-aged and elderly men and women. Journals of Gerontology-Biological Sciences and Medical Sciences, 55(2), B95.

Bottaro, M., Martins, B., Gentil, P., & Wagner, D. (2009). Effects of rest duration between sets of resistance training on acute hormonal responses in trained women. Journal of Science and Medicine in Sport, 12(1), 73-78.

Line 132: Add the inclusion criteria for upper and lower body training.

Line 170: How did you manually recalculated data? Be explicit.

Line 261: The differences in beginners (Figure 3) show no difference between upper and lower limb. This should be also the bullet point of manuscript and in the abstract. This is however, due to only one study, how you want to deal with this in discussion.

Line 324: Does exercise TEMPO influence hormonal response?

Wilk, M., Krzysztofik, M., Petr, M., Zając, A., & Stastny, P. (2020). The slow exercise tempo elicits higher glycolytic and muscle damage but not endocrine response that conventional squat. Neuroendocrinology Letters, 41(5).

Wilk, M., Golas, A., Stastny, P., Nawrocka, M., Krzysztofik, M., & Zajac, A. (2018). Does tempo of resistance exercise impact training volume?. Journal of human kinetics, 62(1), 241-250.

Reviewer #2: General Response

This is an interesting and much needed meta-analysis. It has a few misinterpretations that could need attention.

Specific Responses

Page Line

2 34 It might be helpful to provide a better description of “advanced subjects”. Does that mean “previously trained” or “well-trained”?

2 35 What does “same load” mean?

3 67-69 This does not seem like a complete sentence.

3 79 Should this be “The current review…” to distinguish your work?

3 89 “…strength are increased more than ….”

3 94 What kind of “responses”? Could you be more specific?

3 95 A 25% increase in 15 weeks does not seem to fit the 4.3% per week.

4 109 Why only these two languages?

5 171 “The data from the meta-analyses were …..” in data is a plural word.

14 281 “…frequency per week o number of repetitions….”

14 23 “The data here are very difficult…..”

15 304 “The quality of the studies was considered good….”

15 320 No period after “repetitions”

16 358 Is this a separate paragraph?

16 379 “in addition, it should be noted that Jansen de Jonge, Thompson and Han [15]….”

16 386 “…women, not only are proteins ….”

16 391 LH has not been defined. It should probably be spelled out before using the LH abbreviations.

16 396 There was no reference 15 in the introduction.

17 430 “…studies, since other authors….”

Reviewer #3: Dear Authors,

Manuscript entitled Weekly percent 1RM gains are higher in the lower-body than the upper-body in resistance training experienced healthy young women - a systematic review with meta-analysis about female resistance training

Clarity of content and adequacy to scientific language was demonstrated throughout the manuscript. In addition, the manuscript is interesting, however it is necessary to make some considerations.

#GENERAL CONSIDERATIONS

#I suggest that the authors change the title to Muscle strength gains per week are higher in the lower-body than the upper-body in resistance training experienced healthy young women - a systematic review with meta-analysis

#I suggest that the authors improve the research rationale. Only the scarcity of studies is not enough to justify carrying out this research.

#Methods

#Curiosity, why did the authors not search for the words resisted exercises?

#Although it was mentioned in the limitations, as there are many possibilities to manipulate resistance training variables, how did the authors make this equalization, considering the influence of possibilities on strength gain?

#i.g., different rest intervals between sets and exercises of the studies included in the analysis of this research;

#different volumes (number of sets x number of repetitions x training frequency) of the studies included in the analysis of this research;

#different contraction speeds, as well as range of motion of the studies included in the analysis of this research.

#Discussion

#I suggest that authors begin with the main research findings.

#I suggest that the authors clearly demonstrate the applicability of the research findings.

authors clearly demonstrate the applicability of the research findings.

#Conclusion

#While the findings are interesting, I am not sure the authors should claim the following.

The upper-body can be trained with lighter weights and more repetitions to increase the 1RM compared to the lower-body. The lower-body should be trained with a maximum of six repetitions per set. The lower-body should be trained two times a week, while in the upper body both two to three times a week resulted in similarly muscle strength gains.

#Further studies are needed to confirm these findings in view of the different characteristics of the resistance exercise protocols.

6. PLOS authors have the option to publish the peer review history of their article (what does this mean?). If published, this will include your full peer review and any attached files.

Reviewer #1: No

Reviewer #2: No

Reviewer #3: No

---

## [Author Response · Author response to Decision Letter 0]

13 Mar 2023

Dear Reviewers,

we have now adapted our manuscript accordingly. .

We want to thank you for the very pleasant and, at the same time very constructive suggestions for improvement. We now hope we can satisfy them with the update version of our Systematic Review.

Best regards from Germany

Roger Jung

Reviewer #1: General comments:

The manuscript: „WEEKLY PERCENT 1RM GAINS ARE HIGHER IN THE 2 LOWER-BODY THAN THE UPPER-BODY IN RESISTANCE 3 TRAINING EXPERIENCED HEALTHY YOUNG WOMEN - A 4 SYSTEMATIC REVIEW WITH META-ANALYSIS ABOUT 5 FEMALE RESISTANCE TRAINING“, deals with real Research gap in sport science. I agree that there is a lack of women’s studies in resistance training, but the authors gathered articles by well-defined criteria.

* Thank you very much for your positive evaluation. We really much appreciate.

There one issue in the conclusion statement. Authors are presenting that lower body gains in RM are higher than upper body. However, the difference 4.3% per week in the upper body and 4.7% per week in the lower-body exercises, does not seems to be relevant to this conclusion. Did you make statistics on this? What are minimal detectable changes for % of RM? (please reference this).

* Thank you very much for your comment. We re-checked our data for the difference between upper and lower body, since the numbers in the text and in the figures were different. The difference between upper and lower body in weekly changes is 2%. Please find the correct numbers now in the text as well.

The introduction of the manuscript very briefly presents basic resistance training variables, where the main parameter is exercise selection. The exercise selection might be whole body training, upper/lower limb training, isolated training etc. Please define what you want to understand as lower body training. Many lower body exercise like deadlifting and squatting, includes a lot of back and trunk muscle involvement, how you can distinguish the upper and lower body training. This should be mentioned in the inclusion criteria and discussion (including muscle group size). The absence of full body training is one of the main limitations of this study, try to mention that in the discussion.

* You are right. Many leg exercises involve more muscles than just leg muscles. Nevertheless, the focus of the chosen exercises is on the lower body. We have added information about our definition for upper- and lower body exercises and in the methods and about the missing analysis of full body adaptations in the discussion. 

The conclusion of the study should be clarified by reporting separately on beginners and advanced resistance-trained women. This should be in the abstract as well since there are. 

* We have now added information about beginners and advanced women in the abstract and the conclusion.

The study discussion should mention all loading parameters of resistance training. Therefore I recommend the article after major revisions.

* We now mention training repetitions per sets, number of sets per workout and training frequency per week in the discussion. Additionally, we have now added important parameters such as rest time between the sets in table 2.

Specific comments:

Line 53: the bullet point “Weekly percentage 1RM gains are higher in training experienced women in lower body rather than in upper-body exercises“ seems to be obvious to add by how much they can strengthen more in lower limbs.

* Thank you for pointing this out. However, the differences are really significant. For advanced students, the difference is 56.7 %. For beginners, the differences are 3.7 % per week. We have classified these differences in our results.

Line 69: What about arterial stiffness?

Jurik, R., Żebrowska, A., & Stastny, P. (2021). Effect of an Acute Resistance Training Bout and Long-Term Resistance Training Program on Arterial Stiffness: A Systematic Review and Meta-Analysis. Journal of clinical medicine, 10(16), 3492.

* Thank you for the comment. We have added the reference and information about arterial stiffness.

In introduction or discussion, try to justify whether hormonal response of man and women really differ and how.

Hakkinen, K., Pakarinen, A., Kraemer, W. J., Newton, R. U., & Alen, M. (2000). Basal concentrations and acute responses of serum hormones and strength development during heavy resistance training in middle-aged and elderly men and women. Journals of Gerontology-Biological Sciences and Medical Sciences, 55(2), B95.

Bottaro, M., Martins, B., Gentil, P., & Wagner, D. (2009). Effects of rest duration between sets of resistance training on acute hormonal responses in trained women. Journal of Science and Medicine in Sport, 12(1), 73-78.

* We are of the opinion that this should only take up a short part. However, we would like to point out that there are findings by Kissow and colleagues (DOI: 10.1007/s40279-022-01679-y) , or rather assumptions, that an estrogen peak at the end of the follicular phase could have a positive effect on strength training success. 

Our study includes only young women (average age: 25.5±3.4 years) before menopause. We do not show differences between men and women in our analysis. But we report in our discussion that there are differences in muscle fibre distribution between women and men and therefore differences in training control and results.

We have added Battaro (DOI: 10.1016/j.jsams.2007.10.013 ) and thank you for pointing it out as it includes young women. We address this here in our limitations.

Line 132: Add the inclusion criteria for upper and lower body training

* We have added information here. We added a list of upper- and lower-body exercises in the methods and the results.

Line 170: How did you manually recalculated data? Be explicit.

* We calculated manually using a calculator. To avoid confusion we deleted this here.

Line 261: The differences in beginners (Figure 3) show no difference between upper and lower limb. This should be also the bullet point of manuscript and in the abstract. This is however, due to only one study, how you want to deal with this in discussion.

* We included this in the abstract and the bullet points in the manuscript. We have mentioned the statistical changes when this one study is removed from the analysis. Since it is a study we could not exclude it will finally influence our results.

Line 324: Does exercise TEMPO influence hormonal response?

Wilk, M., Krzysztofik, M., Petr, M., Zając, A., & Stastny, P. (2020). The slow exercise tempo elicits higher glycolytic and muscle damage but not endocrine response that conventional squat. Neuroendocrinology Letters, 41(5).

Wilk, M., Golas, A., Stastny, P., Nawrocka, M., Krzysztofik, M., & Zajac, A. (2018). Does tempo of resistance exercise impact training volume? Journal of human kinetics, 62(1), 241-250.

* Obviously, exercise tempo does have an influence on training adaptations. Nevertheless, this was not focus in our review. Therefore, we did not analyse the studies with regards to tempo. But have now included this in our limitations.

Reviewer #2: General Response

This is an interesting and much needed meta-analysis. It has a few misinterpretations that could need attention.

Specific Responses

Page Line

2 34 It might be helpful to provide a better description of “advanced subjects”. Does that mean “previously trained” or “well-trained”?

For the evaluation we used the classification offered by Santos Junior et al. (DOI: 10.1519/SSC.0000000000000627). We have described this in out methods. However, since some studies only used machine exercises, we had to trust the assessments of the authors of the individual studies about the classification of the participants. This is listed accordingly in the limitations. Nevertheless, we think that we were able to integrate a good standard here through the assessment. Due to the complex explanation we refrained from a detailed description in the abstract.

2 35 What does “same load” mean?

* Due to several changes in the abstract we do not have enough space to define training load here. We have added more information about the definition of training load later in the manuscript (cf. results 3.4.1).

3 67-69 This does not seem like a complete sentence.

* You are right. We corrected the sentence.

3 79 Should this be “The current review…” to distinguish your work?

* You are right. Thank you for the comment.

3 89 “…strength are increased more than ….”

* Corrected

3 94 What kind of “responses”? Could you be more specific?

* We have added more information here. 

3 95 A 25% increase in 15 weeks does not seem to fit the 4.3% per week.

* This is true. However, we include other studies than Hagstrom et al.  (DOI: 10.1007/s40279-019-01247-x). For example, we excluded studies in which we can assume that the women have hormone problems due to being overweight, obese, underweight, or other diseases that could influence the endocrine system. Since, Kissow et al. (DOI: 10.1007/s40279-022-01679-y) assume that estrogen potentially has an influence on larger gains in the lower body, we assume that the differences here could be higher. We also use a different database query than these authors.

4 109 Why only these two languages?

* We decided to perform the literature search in the important English databases. In our team the only other language besides English is German. Therefore, we focused on these two languages.

5 171 “The data from the meta-analyses were …..” in data is a plural word.

* Corrected

14 281 “…frequency per week o number of repetitions….”

* Corrected

14 293 “The data here are very difficult…..”

* Corrected

15 304 “The quality of the studies was considered good….”

* Corrected

15 320 No period after “repetitions”

* Corrected

16 358 Is this a separate paragraph?

* Corrected

16 379 “in addition, it should be noted that Jansen de Jonge, Thompson and Han [15]….”

* Corrected

16 386 “…women, not only are proteins ….”

* Corrected

16 391 LH has not been defined. It should probably be spelled out before using the LH abbreviations.

* Corrected

16 396 There was no reference 15 in the introduction.

* Corrected

17 430 “…studies, since other authors….”

* Corrected

Reviewer #3:

Dear Authors,

Manuscript entitled Weekly percent 1RM gains are higher in the lower-body than the upper-body in resistance training experienced healthy young women - a systematic review with meta-analysis about female resistance training.

Clarity of content and adequacy to scientific language was demonstrated throughout the manuscript. In addition, the manuscript is interesting, however it is necessary to make some considerations.

#GENERAL CONSIDERATIONS

#I suggest that the authors change the title to Muscle strength gains per week are higher in the lower-body than the upper-body in resistance training experienced healthy young women - a systematic review with meta-analysis.

* Thank you for the comment. Changed as suggested.

#I suggest that the authors improve the research rationale. Only the scarcity of studies is not enough to justify carrying out this research.

*Thank you for your comment. We have now added more details to the rational of this review. 

#Methods

#Curiosity, why did the authors not search for the words resisted exercises?

* We use the terms “strength training” and “resistance training” we did not not want to include studies that performed strength training with a rubber or resistance band for example. At the beginning of our literature search, we conducted various test queries via PubMed and then finally defined the search string. Furthermore, we did not find more studies with the words “resisted exercise” which were in the focus of our research question. 

#Although it was mentioned in the limitations, as there are many possibilities to manipulate resistance training variables, how did the authors make this equalization, considering the influence of possibilities on strength gain? #i.g., different rest intervals between sets and exercises of the studies included in the analysis of this research; #different volumes (number of sets x number of repetitions x training frequency) of the studies included in the analysis of this research; #different contraction speeds, as well as range of motion of the studies included in the analysis of this research.

* We added now more available data about the training protocol for each included study in table 2. The new information will not influence our results or the figures. The rest times, which we also now added to Table 2, do not have any significant influence either. But we add it in our limitation. In order to ensure a comparison between the studies, we focus on variables that are included in as many studies as possible and thus can be compared with each other. This is therefore the basis for this review. Furthermore, we have also indicated in Table 2 whether or not training was done to muscular failure. We discuss this in the limitations, as we find it questionable whether training with beginners in particular should actually be carried out into muscular failure or whether these types of subjects have a feeling for it.

#Discussion

#I suggest that authors begin with the main research findings.

* Thank you very much. We changed the beginning of the discussion.

#I suggest that the authors clearly demonstrate the applicability of the research findings.

authors clearly demonstrate the applicability of the research findings.

* We added information in the discussion.

#Conclusion

#While the findings are interesting, I am not sure the authors should claim the following.

The upper-body can be trained with lighter weights and more repetitions to increase the 1RM compared to the lower-body. The lower-body should be trained with a maximum of six repetitions per set. The lower-body should be trained two times a week, while in the upper body both two to three times a week resulted in similarly muscle strength gains.

* We have adjusted the wording to make clear that these recommendations are just a result from the studies we analyzed.

#Further studies are needed to confirm these findings in view of the different characteristics of the resistance exercise protocols.

* We also wanted our review to provide an impetus for further research in this area. In addition to our review, we are currently in the process of developing a study to clarify the ambiguities. We rephrase the sentence.

---

## [Decision Letter · Decision Letter 1]

27 Mar 2023

Muscle strength gains per week are higher in the lower-body than the upper-body in resistance training experienced healthy young women - a systematic review with meta-analysis

PONE-D-22-24644R1

Dear Dr. Jung,

We’re pleased to inform you that your manuscript has been judged scientifically suitable for publication and will be formally accepted for publication once it meets all outstanding technical requirements.

Kind regards,

Hans-Peter Kubis, PD. Dr. rer. nat.

Academic Editor

PLOS ONE

Additional Editor Comments (optional):

Reviewers' comments:

Reviewer's Responses to Questions

**Comments to the Author**

1. If the authors have adequately addressed your comments raised in a previous round of review and you feel that this manuscript is now acceptable for publication, you may indicate that here to bypass the “Comments to the Author” section, enter your conflict of interest statement in the “Confidential to Editor” section, and submit your "Accept" recommendation.

Reviewer #1: All comments have been addressed

Reviewer #2: All comments have been addressed

2. Is the manuscript technically sound, and do the data support the conclusions?

Reviewer #1: Yes

Reviewer #2: Yes

3. Has the statistical analysis been performed appropriately and rigorously? 

Reviewer #1: Yes

Reviewer #2: Yes

4. Have the authors made all data underlying the findings in their manuscript fully available?

Reviewer #1: Yes

Reviewer #2: Yes

5. Is the manuscript presented in an intelligible fashion and written in standard English?

Reviewer #1: Yes

Reviewer #2: Yes

6. Review Comments to the Author

Reviewer #1: The authors made apropriate changes in their manuscript. I have only one more minor note, that limitations on the line 432 should be referenced, as it is rest interval referenced in following sentence.

Reviewer #2: The study is an interesting and timely work. It should be useful to those working in the field of resistance training.

7. PLOS authors have the option to publish the peer review history of their article (what does this mean?). If published, this will include your full peer review and any attached files.

Reviewer #1: No

Reviewer #2: No

---

## [Editor Report · Acceptance letter]

31 Mar 2023

PONE-D-22-24644R1 

Muscle strength gains per week are higher in the lower-body than the upper-body in resistance training experienced healthy young women - a systematic review with meta-analysis 

Dear Dr. Jung:

I'm pleased to inform you that your manuscript has been deemed suitable for publication in PLOS ONE. Congratulations! Your manuscript is now with our production department. 

Kind regards, 

on behalf of

Dr. Hans-Peter Kubis 

Academic Editor

PLOS ONE